# Evolution of Electrochemical Impedance Spectra Characteristics of Cementitious Materials after Capturing Carbon Dioxide

**Qiong Liu** [1], **Huilin Tang** [1], **Lin Chi** [1,*], **Kailun Chen** [1,*], **Lei Zhang** [2] and **Chaoxiong Lu** [2]

1   School of Environment and Architecture, University of Shanghai for Science and Technology, Shanghai 200093, China
2   Hebei Expressway Jingxiong Preparation Office, Baoding 071000, China
*   Correspondence: chilin@usst.edu.cn (L.C.); 183811838@st.usst.edu.cn (K.C.)

**Abstract:** The electrochemical parameters of cement-based materials with different water–cement ratios in carbon curing and water curing were measured with electrochemical impedance spectroscopy (EIS). The optimized circuit model and corresponding electrical parameters were obtained to illustrate the variation of the microstructure of cementitious materials after carbon capturing. The results show that, to a large extent, the semicircle diameter in the high frequency area gradually increased along with carbon curing and water curing. However, carbon curing showed a difference that the semicircle diameter in the high frequency appeared at the minimal value at 3 days, which was higher than that at 1 day and 7 days. This should be the result of the joint influence of water content and porosity in the cement matrix. It was also found that the mass increase rates of carbonation with water–cement ratios of 0.4, 0.5, and 0.6 were basically stable at 3.4%, 5.0%, and 5.5%, respectively. The electrochemical parameters $\rho_{ct2}$ of cement mortar corresponding to carbon curing were around three times that of water curing specimens, mainly due to the reduction of soluble materials and refinement of connecting pores in the microstructure of cementitious materials. A quadratic function correlation between the mass increase rate and $\rho_{ct2}$ in the carbonation process of cement mortar was built, which proved that EIS analysis could be applied to monitor the carbon capturing of cement-based materials, either for newly mixed concrete or for recycled concrete aggregates.

**Keywords:** carbon capturing; electrochemical impedance spectroscopy; cement-based materials; equivalent circuit model; recycled concrete aggregates

## 1. Introduction

Cement-based materials are ubiquitously applied in infrastructure projects due to their excellent properties, such as fine strength and plasticity [1]. With the rapid development of construction and building trade, the carbon footprint in cement fields is one of the main reasons for global warming and the Antarctic ozone hole, which in turn leaves our living environment dilemmatic. At present, China is the first in carbon emissions in this world. From 2004 to 2019, the carbon emissions of China's construction industries increased from about 1.5 billion tons to nearly 5 billion tons [2]. As an environmental advocate in international communities, China has vigorously implemented the "Carbon peaking, carbon neutrality" policy and continuously explored the way of low carbon.

Studies have reported that cement-based materials can solidify $CO_2$ so that it enters the inner structures of cement to form stable substances. In addition, it can improve the compactness and strength of cement-based materials [3]. In recent years, research related to the solidification of $CO_2$ has been conducted. There are substantial differences between natural carbonation and cement's one; $CO_2$ can affect cement-based materials to generate stable crystals inside and then form a bonding grid structure. Some gel products, such as dicalcium silicate and tricalcium silicate, would fill the internal pores of materials, improve the compactness of the microstructure, and enhance the strength and durability of cement-based materials [4,5].

Optimizing the microstructure of cement-based materials in carbonation is the premise to study the behavior of carbonation in concrete and to probe into its mechanisms [6]. Carbonation in concrete materials can be equal to the photosynthesis in flora, which is regarded as an eco-friendly practice, referring to the reaction between $CO_2$ and alkaline carbonaceous substances in cement-based materials, thus helping reduce the amount of $CO_2$. After carbonation, some hydrated products can be found: calcium hydroxide (CH), carbides, hydrated calcium silicate gel (C-S-H), calcium alum (AFt), low-sulfur hydrated calcium sulfoaluminate (AFm), and even not hydrated cement phases (such as tricalcium silicate, dicalcium silicate, etc.) [7]. The reaction of these substances changes the pH value and the internal structure of cement hydration products [8]. In this techno-savvy era, variedly advanced analysis technologies have been used to study the carbonation process of cement-based material [4]. Shi et al. [9,10] have shown that with the increase of curing time in the carbonation curing process, the crystallinity of $CaCO_3$ crystals inside cement-based materials is proportional to the density of the internal structure. Klemm et al. [11,12] have shown that many factors affect the variation of the microstructure in the carbonation process in cementitious materials, among which the water–cement ratio is the most significant. With the increase of the water–cement ratio, the degree of carbonation increases. However, a too-high water–cement ratio would lead to a decrease in the amount of hydration product, which may have an impact on the capability of $CO_2$ sequestration and the mechanical strength of cement-based materials.

Electrochemical impedance spectroscopy (EIS) can monitor the transformation of the microstructure in cement-based materials without damage [13]. The test devices are composed of concrete specimens, an electrode, an electrochemical workstation, and electrolyte, and the workstation provides an excitation current at different frequencies [14,15]. Electrical signals are transmitted through the electrode and specimens connected to the electrolyte to obtain electrical responses about the ion concentration and microstructure of the concrete [16,17]. Ion concentration in cement-based materials is affected by hydration and carbonation [18]. In the hydration process, the dissolution or reaction of cement particles in the matrix is conducive to unblocking the migration path of ions, while the hydration products hinder the migration of ions, forcing them to change the migration direction and extend their path, thus leaving the migration difficult. The speed of dissolving cement granules and generating hydrated products significantly influences the variation of electro-response [14,19]. In the carbonation phase, although there is the consumption of hydration products, the volume of carbonation products increases under the consumption of equimolar substances, and the products attach to the pore structure, making the pore structure gradually dense. This process also affects the migration and transmission of ions and ultimately affects the electrical response of cement-based materials [20,21].

To make a better thorough inquiry into the transformation of cement-based materials' microstructure under electrochemical conditions, cementitious materials can be divided into three phases: solid phase, liquid phase, and solid–liquid interface, and the ion contents and its double-layer structures may be altered with the changes of ion and water molecules vibrating under the premise of alternating current's influence, which may lead to changes in impedance [22,23]. Since the microstructure, as the platform, can reflect the degree of carbonation in cement-based materials, there is a difference in the molar mass of reactants and products [24]. The study of EIS in the carbonation process of cement-based materials is relatively blank in current research, and the mechanism of the microstructure in cement-based materials caused by carbonation is not yet clear.

Our aims were to use EIS to characterize the electrical properties of cement-based materials in carbon curing for better understanding of the carbon-capturing process of newly mixed concrete and recycled concrete aggregates. The influence of carbon curing age and water–cement ratio on the carbonation degree, as well as the microstructure change of cement-based materials with mercury intrusion porosimetry (MIP), was explored. Through setting an electrochemical circuit model, the relationship within three sectors: electrochemical parameters, the carbonation degree, and microstructure of cement-based materials

could be coined, which may carve the path for future studies in the electrochemical characterization method.

## 2. Materials and Methods

### 2.1. Material and Specimen Preparation

The cement used in this study was P.O. 42.5 ordinary Portland cement, and its chemical composition is shown in Table 1. The fine aggregates were from natural river sand with particle size less than 0.6 mm.

**Table 1.** Chemical composition of cement (w/%).

| CaO | $SiO_2$ | MgO | $SO_3$ | $Al_2O_3$ | $K_2O$ | $Na_2O$ | $Fe_2O_3$ | LOI |
|---|---|---|---|---|---|---|---|---|
| 63.8 | 20.78 | 1.72 | 3.82 | 3.57 | 0.75 | 0.26 | 3.99 | 1.31 |

The mixing proportions of the cement mortar with different water–cement ratios (0.4, 0.5, and 0.6, respectively) and the same sand–cement ratio (3:1) are shown in Table 2. The specimens were made in 20 mm × 20 mm cylinders for carbon curing and water curing. Each curing method contained three groups of specimens with different water–cement ratios, and each group had three specimens. In Table 2, the letters "C" and "W" in specimen names represent carbon curing and water curing, respectively. After this letter, the number represents the water–cement ratio.

**Table 2.** Mix proportions.

| Specimen | Cement (g) | Sand (g) | Water (g) |
|---|---|---|---|
| C4 and W4 | 100 | 300 | 40 |
| C5 and W5 | 100 | 300 | 50 |
| C6 and W6 | 100 | 300 | 60 |

First, the cement and sand were mixed for 30 s; then, water was added with continuous stirring for another 30 s. After pouring the mixed mortar into molds, they were vibrated until no bubbles floated up from the mortar. A copper wire with a diameter of 1.1 mm was buried into the mortar to a depth of half the height of the specimens. After 24 h of curing, the specimens were demolded and sent to a carbon-curing chamber with normal curing conditions.

### 2.2. Test Method

The carbon-curing group was for a simulation of newly mixed concrete or recycled concrete aggregates in the carbon-capturing process. After demolding, the specimens with copper wire electrodes were placed in the carbon-curing chamber and the concrete standard curing chamber. The parameters of carbon curing were as follows: relative humidity 70 ± 2%, $CO_2$ concentration 20 ± 2%, temperature 20 ± 2 °C, and pressure 2 bar. The parameters of standard curing were: relative humidity 98 ± 2% and temperature 20 ± 2 °C. It is noteworthy that specimens in the carbonation group were placed in a plastic box with a layer of saturated sponge placed at the bottom of the box to ensure the hydration of cement. Small 1 mm holes were drilled on the box cover to let the $CO_2$ flow and keep the $CO_2$ concentration in the box stable at about 20%. The EIS of specimens was tested at the curing ages of 1, 3, 7, 14, 28, and 56th days. The flow chart of this study is shown in Figure 1.

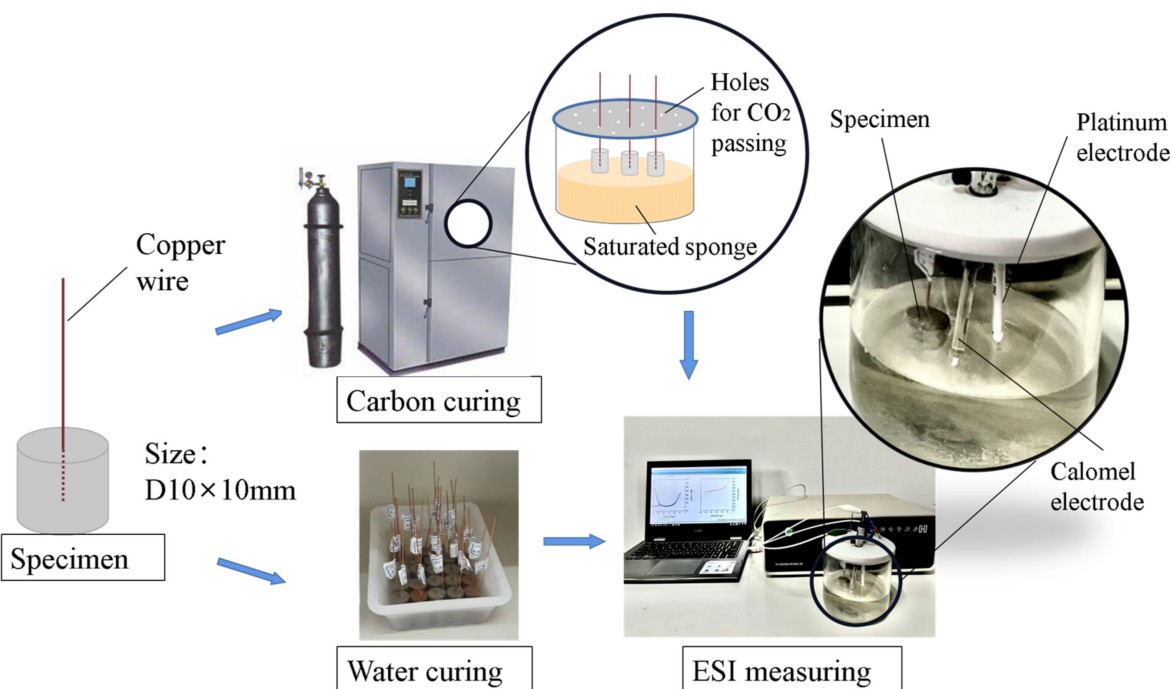

**Figure 1.** Specimen-curing environment and EIS measurement process.

For electrochemical AC impedance, we used a CHI604E electrochemical workstation produced by Shanghai Chenhua Instrument Co., Ltd., Shanghai, China. The test environment was saturated with a $Ca(OH)_2$ (CH) solution. The three-electrode test method was used, and a platinum sheet was the counter electrode, a saturated calomel electrode was the reference electrode, and the specimen was the working electrode. The test frequency was set at 1 Hz–1 MHz. Before each test, the specimens were immersed in a saturated CH solution for 1 h to keep the specimens in a saturated state.

For the carbonation-group specimens, the mass-increase method was used to calculate the content of $CO_2$ absorption [25]. Specifically, the specimens were placed in a drying oven ($103 \pm 2$ °C) until a constant weight and then weighed to obtain the mass increase ratio. In addition, the same specimens were made and placed in a concrete standard curing chamber for water curing, and the mass increasing rate of the hydration process was determined by drying and weighing as well. The difference between the above rates was the carbonation weight gain rate, that is, the amount of $CO_2$ absorption.

For weight gain by baking to absolute drying, the formula was as follows:

$$\varphi = \frac{m_a - m_b}{m_a} \tag{1}$$

where $\varphi$—weight gain rate; $m_a$, $m_b$—mass of specimens before and after carbon curing or water curing.

X-ray diffraction (XRD) and MIP were used to study the reaction mechanism of $CO_2$ capture by cementitious materials. This further supports the evolution of electrochemical impedance spectroscopy before and after the carbonation reaction.

## 3. Results

### 3.1. Electrochemical Impedance Analysis

Figure 2 shows the EIS curve of each group of specimens with the increase of carbon curing age. The real part and imaginary part of impedance are characterized by resistivity, and the following equation was used for conversion between resistivity and resistance:

$$R = \rho L / S \tag{2}$$

where $R$ is the resistance, $\Omega$, $\rho$ is the resistivity of the test specimen, $\Omega \cdot cm$, $L$ is the height of the cylinder specimen, cm, $S$ is the circular cross-section area of the specimen, $cm^2$.

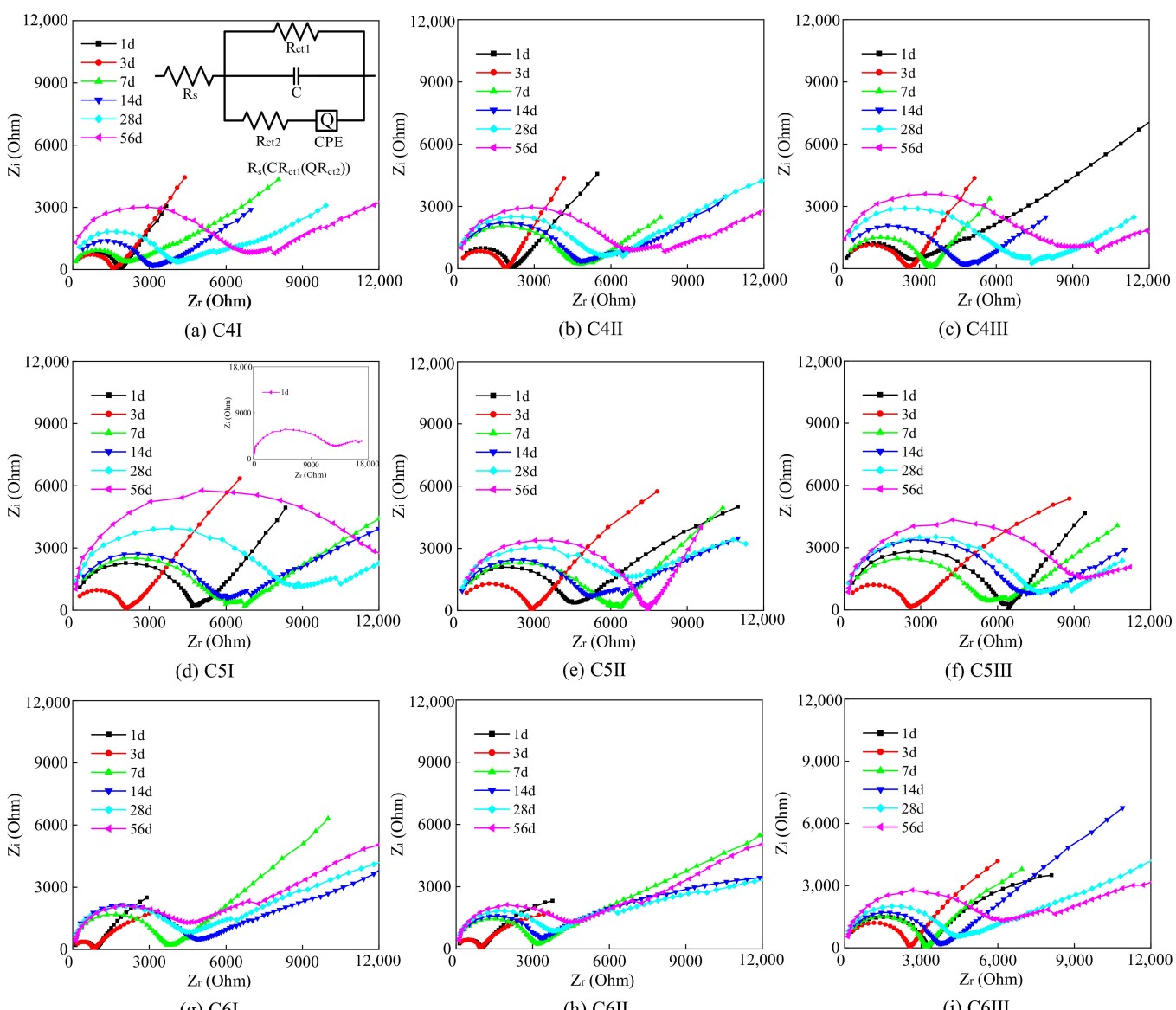

**Figure 2.** EIS of specimens in carbonation curing: (**a**) C4I; (**b**) C4II; (**c**) C4III; (**d**) C5I; (**e**) C5II; (**f**) C5III; (**g**) C6I; (**h**) C6II; (**i**) C6III.

The impedance spectroscopy curves are composed of a semicircular arc and a straight line, in which the high frequency region is the circular arc, and the low-frequency region is the straight line. The starting point of the high frequency arc approximates the zero point, which is consistent with the characteristics of the impedance spectroscopy curve of cement-based composites [26].

In order to further obtain the characteristic parameters of impedance spectroscopy, many scholars have described the equivalent circuit of the electrochemical testing cement matrix [27] and used various characteristic parameters to characterize the micropore structure, pore solution concentration, ion composition, etc. In general, as curing age increases, there is an increase in the arc diameter in the high frequency area of impedance spec-

troscopy and the corresponding imaginary position (i.e., ordinate) of the inflection point between the arc in the high frequency area and the straight line in the low-frequency area, but there is a corresponding drop in the slope of the straight line in the low-frequency area. This is mainly because with the deepening of carbonation, the microstructure of the cement matrix gradually becomes dense, the connected ionic conductivity path decreases significantly [28], and the resistivity of the specimen gradually increases.

It is worth noting that the diameter of the high frequency arc of cement mortar increases with the increase of carbonation time, but almost all specimens presented a phenomenon: the high frequency arc of the third day was located in the interior of the first day, which indicated that the resistivity of the specimen was the lowest when carbonation was at the third day. This may have been because in the process of the carbonation hydration reaction, when the curing age was three days, the $CO_2$ had not completely penetrated into the cement-based material, and the content of soluble salts in the specimen was high [29]. Therefore, the impedance spectroscopy curve obtained from the test showed the smallest high frequency arc.

According to the impedance spectroscopy curve of the 28th day results, when the water–cement ratio was 0.4, the high frequency arc of the impedance spectroscopy curve was the smallest, and the average diameter was about 5 k$\Omega$; when the water–cement ratio was 0.5, the high-frequency arc was larger, and the average diameter could reach 11 k$\Omega$; when the water–cement ratio was 0.6, the average diameter of high frequency arc was about 6 k$\Omega$.

For the EIS test, the total pore volume of the specimens, especially the number of connected pores, had a great influence on the testing results. The more connected holes, the lower the resistivity; in addition, impedance spectroscopy was also related to the content of free mobile ions in the pores. The resistivity of specimens with more free ions is lower [30], which is due to the mutual balance of these two effects. In general, the higher the water–cement ratio, the lower the resistivity [31]. Additionally, the lower the water–cement ratio, the smaller the pore volume of the specimen, which inhibits the progress of carbonation and retains more free-moving ions, and the lower the measured resistivity. All these together cause the resistivity of the specimens in the carbonation group to increase first and then decrease with the increase of the water–cement ratio.

Figure 3 shows the impedance spectroscopy curves of each group's specimens in water curing. Firstly, it can be found that the high frequency arc of cement mortar was the smallest at the first-day hydration. With the increase of hydration age, the high frequency arc of cement mortar gradually became larger. Moreover, most of specimens also showed that the inflection point position between the high frequency arc and the low-frequency straight line gradually increased, and the slope of the low-frequency straight line decreased. As the process of hydration was developing constantly and the products it generated were filled in the pores, ion concentration in the pore solution had fallen, which led to the upward trend of resistivity. In addition, with the progress of hydration, the porosity gradually decreases, which is also the reason why the resistivity of cement-based composites gradually increases. Regarding different water–cement ratios, the resistivity of the specimen under hydration curing showed that it decreased with the increase of the water–cement ratio, which was in line with the definition: the greater the water–cement ratio, the greater the porosity of cement-based materials, and the smaller the resistivity [30]. In addition, the maximal change of the impedance spectroscopy curve of hydration-curing specimens with curing age was from 28 days to 56 days, indicating that the microstructure of cement-based materials had the largest change during this period, and earlier hydration products crystallized into a more stable state and filled the internal pores during this period. Compared with the hydration-curing condition, the impedance spectroscopy curve of the carbonized curing specimens changed greatly from the third day to the seventh day.

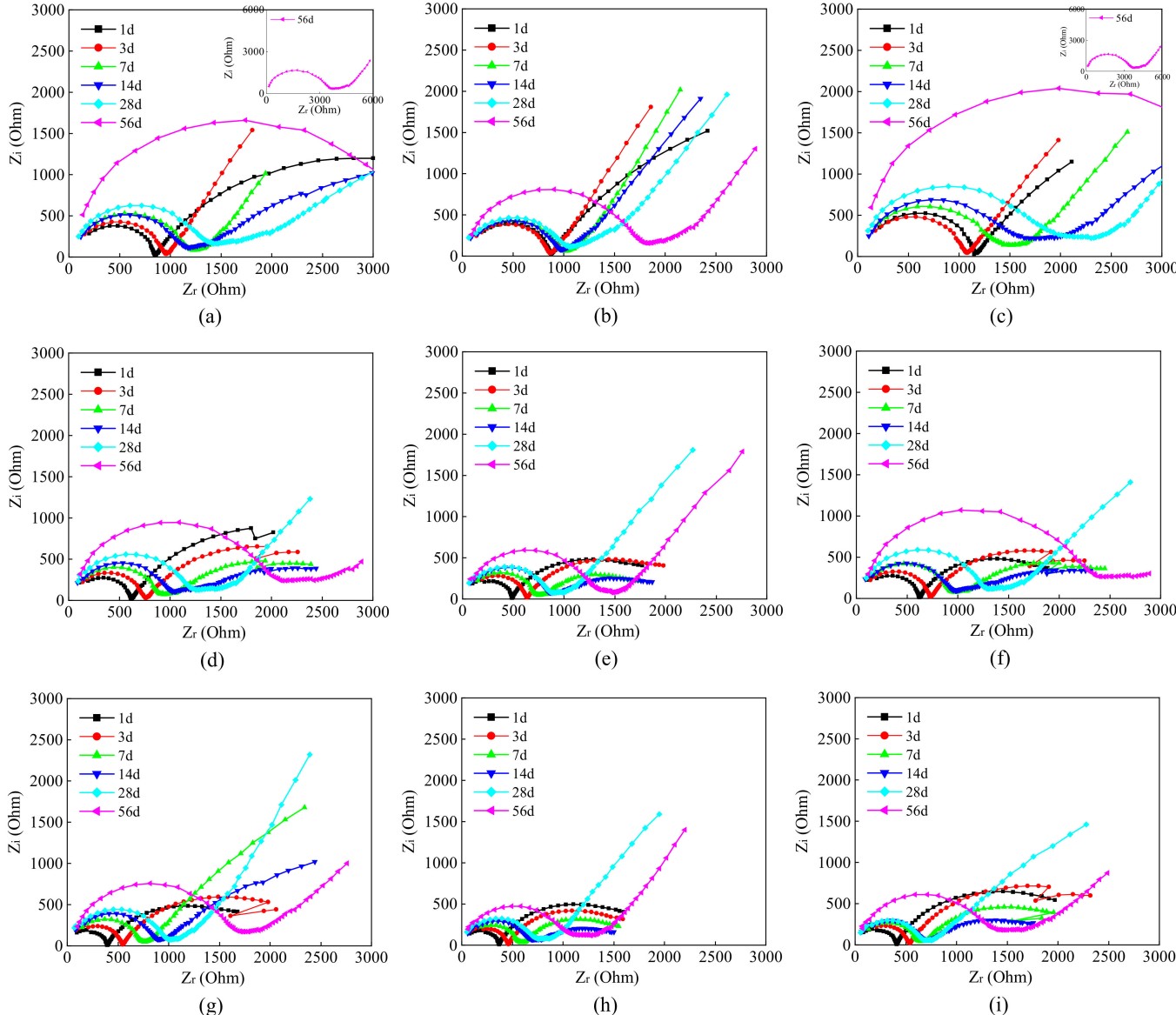

**Figure 3.** EIS of specimens in water curing: (**a**) W4I; (**b**) W4II; (**c**) W4III; (**d**) W5I; (**e**) W5II; (**f**) W5III; (**g**) W6I; (**h**) W6II; (**i**) W6III.app.

Generally, in the ordinary Nyquist curve, the half arc applicable in the high frequency region represents the resistance or resistivity of ions in the cement matrix microstructure, including the liquid phase and solid phase, while the second arc in the low-frequency region reflects the interface behavior between the electrode and the cement matrix [32].

### 3.2. $CO_2$ Absorption

Figure 4 represents the mass increase rate of the specimens in carbonation curing. It can be seen that the mass increase rate of each specimen showed a fast upward trend and then slowed with the mass increase rate 35% in 1 day, 50% in 3 days, 75% in 7 days, and 95% in 14 days. For cement-based composite materials which are still in the early stage of hydration, the carbonation reaction mainly includes three types: the reactions between unhydrated cement particles and $CO_2$, $CO_2$ participating in the hydration reaction, and reactions of hydration products CH and C-S-H with $CO_2$. Among them, the rapid mass increasing of the early curing age is mainly due to the high content of CH in the early reaction products of cement, which preferentially participate in the carbonation reaction.

There is a large difference in the relative molar mass between the reactants and products, which leads to significant mass growth. With the progress of carbonation and hydration, the mass tends to be stable. On the one hand, $CaCO_3$ generated by the reaction refines the internal pores to prevent the reaction from continuing. On the other hand, when CH is consumed, $CO_2$ mainly reacts with substances such as C-S-H. The relative molar mass of these reactants and products is similar, therefore, the mass growth of the test pieces is not obvious. It was illustrated that the former is the main reason [33].

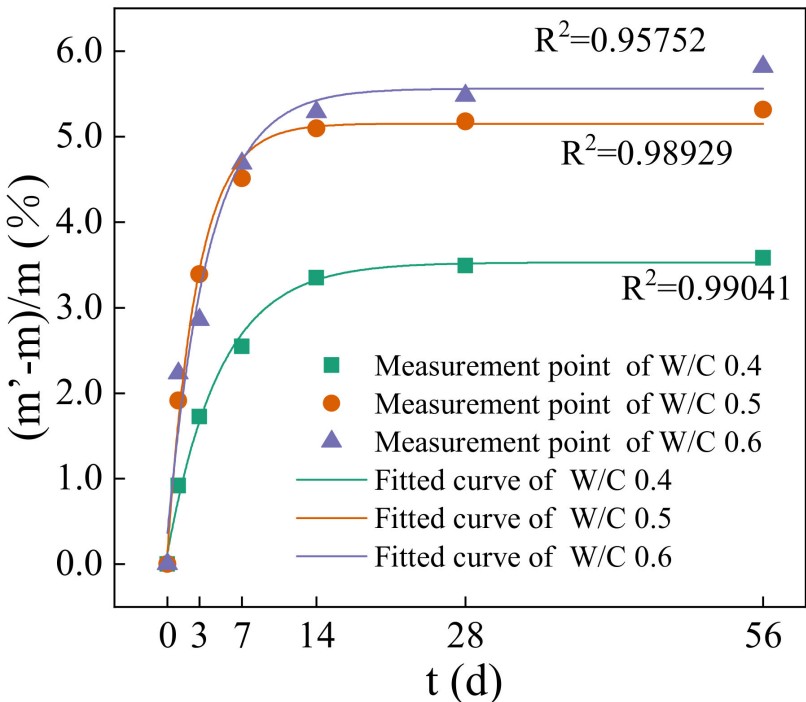

**Figure 4.** The rate of mass change in the carbonation process.

In the carbonation curing process of cement-based materials, the two basic reactions of carbonation and hydration develop in a coupling way. The evolution of the specimens' microstructure is mainly determined by the ratio of dynamics in the two reaction processes [34]. According to the test of weight-increasing rate in the hydration process, the mass-increasing rates of the three kinds of water–cement ratio specimens were about 0.5%, 1.0%, and 1.5% after 3, 7, and 14 days of hydration curing, and then tended to be stable. According to the literature [35], the hydration degree of cement paste can reach 80% on the 7th day, after which the hydration slows down and starts to stabilize. The increase of mass change on the 7th to 21st day had no obvious change compared with that on the 7th day. In Figure 4, after 14 days of carbon curing, the mass increase rates of specimens with water–cement ratios of 0.4, 0.5, and 0.6 were basically stable at 3.4%, 5.0%, and 5.5%, respectively. It can be seen from the comparison that the mass increase rates of carbonation were 1.9%, 3.5%, and 4%, respectively. Of course, the carbonation reaction and hydration reaction cannot be completely split, and the mass increase rate cannot be simply considered as the carbonation mass increase rate. Even in a dry environment, the carbonation reaction process itself will generate water, which provides the demand for the hydration reaction [18].

Regarding different water–cement ratios, the lower the water–cement ratio, the smaller the carbonation mass increasing rate. When the water–cement ratio was 0.4, the maximal mass increase rate of each specimen was about 3.5%, and when the water–cement ratio was 0.5 and 0.6, it was 5.2% and 5.7%, respectively. This is because with the increase of

the water–cement ratio, the porosity of cement-based materials increases after cement hydration, which is conducive to the diffusion, absorption and carbonation of $CO_2$ [36].

The curve in Figure 4 is a function curve with fitting degree above 0.95, and the fitting function is taken as:

$$y = a - be^{-kx} \tag{3}$$

In the corresponding functions of the three fitting curves, $a$, $b$, and $k$ can be gained with positive values, which are shown in Table 3. The value of $a$ determines the value of the final trend value of the function and indicates the value of the mass increase rate of carbonation after completion. The greater the absolute value of $b$ and $k$, the smaller the function value, which shows that $b$ and $k$ jointly determine the change speed of the mass-increasing rate at the initial stage of carbon curing.

**Table 3.** Fitting parameters of carbonation process curves.

|  | $a$ | $b$ | $k$ |
| --- | --- | --- | --- |
| 0.4 | 3.528 | 3.393 | 0.201 |
| 0.5 | 5.152 | 5.004 | 0.357 |
| 0.6 | 5.562 | 5.194 | 0.259 |

*3.3. MIP*

The pore structure of cement paste includes gel pore (0.5–10 nm), capillary pore (10–1000 nm), and macropore (1000–5000 nm). Among them, the capillary pores are usually determined by the water–cement ratio. For the capillary micropores of 10–1000 nm, the pore size and volume content are of great significance in controlling the compressive strength of cement paste [37]. Since the strength of cement-based materials mainly depends on capillary pores and macropores, but rarely on gel pores, and capillary pores can generally form a continuous grid in the cementitious materials and connect with the outside smoothly, the MIP method is used to determine the pore structure of cement-based materials with reliability [38].

Figure 5 shows the MIP curve of each group of specimens after 56 days in carbon curing. It can be found that the average size of micropores and macropores increased with the rising of the water–cement ratio and decreased with the drop of curing age. The total porosity of the C5 group was the lowest, which may have resulted from the carbon-curing effect being the best for C5. The volume content of micropores increases with curing time, indicating that some macropores are filled and transformed into micropores [39]. This result is consistent with the phenomenon of resistivity with the water–cement ratio in carbonation curing in impedance analysis. In the histogram of Figure 5b, the porosity and percentage of pores smaller than 100 nm increased, while the porosity of pores larger than 1000 nm decreased [40], indicating that the carbonation reaction takes place preferentially in macropores or larger pores, and carbonation products deposit in large pores to block the pores, or large pores are divided into small pores. It was also studied in the literature [41] that for cement paste after carbonation, the structure of macropores has not changed significantly, while the volume of macropores in the paste mixed with other cementitious materials has increased significantly. This may be due to the fact that the number of macropores in the paste is small, and the carbonation reaction is more concentrated in the pores, so the consumption of macropores is significantly increased [42,43], and the volume content of macropores is in relative dynamic equilibrium.

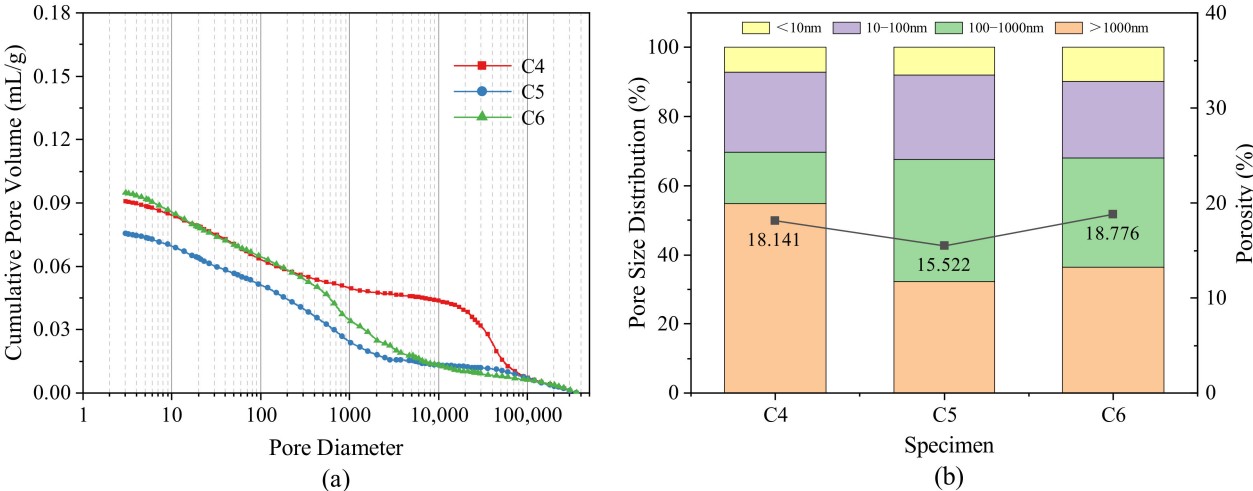

(a)

(b)

**Figure 5.** Cumulative pore volume and pore structure distribution of C4, C5, and C6 groups after curing for 56 days: (**a**) cumulative pore volume curve; (**b**) pore structure distribution.

### 3.4. X-ray Diffraction

XRD detection can study the phase composition of materials, and the XRD spectra that simulate cementitious materials to capture $CO_2$ were collected as shown in Figure 6. Comparing the uncarbonated sample with the 56-days-carbonated sample, the diffraction peaks of portlandite (2θ at about 18° and 34°) were observed in the XRD patterns of the uncarbonated one, as expected. For the carbonated sample, the peaks of calcium hydroxide decreased significantly. In addition, microcline and calcite were detected at about 26° and 29°, respectively, for the uncarbonated one. However, the peak height of these crystals was low, indicating that the sample had been carbonized during making, and the carbonation process was not high [44]. Another obvious difference is that the carbonized specimen generated ore crystals such as calcite and vaterite, indicating that under long-term carbonation curing, the carbonation products converted from the main calcium carbonate calcite into a more stable crystal state, vaterite, which is conducive to a denser microstructure [45].

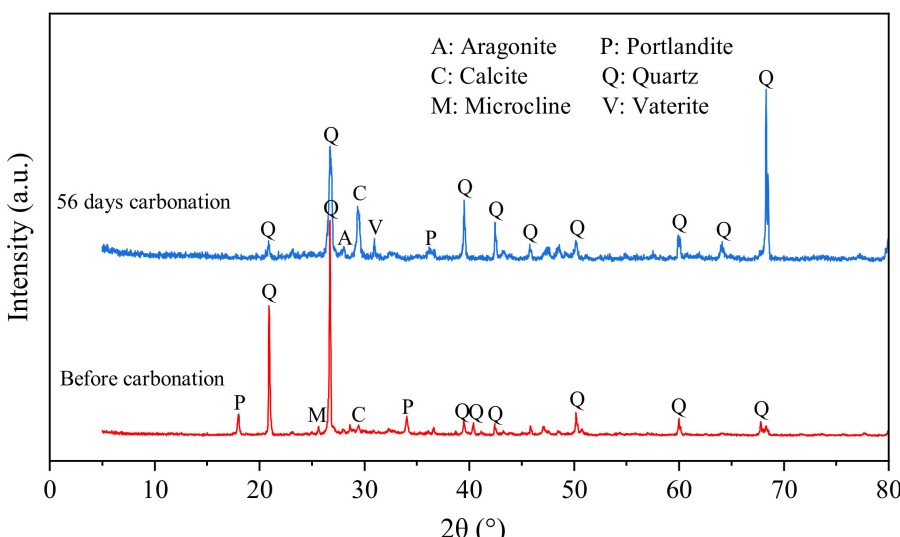

**Figure 6.** XRD results of sample before and after carbonation curing.

## 4. Discussion

### 4.1. Selection of Equivalent Circuit Model

The use of EIS to characterize the physical and chemical changes of carbonation curing of cement-based materials should be based on the selection of suitable equivalent circuits to obtain electrochemical parameters. At present, there are plenty of models for the electrochemical equivalent circuit of cement-based material systems, for example, the Randles equivalent circuit model $R_s(Q(R_{ct}W))$ proposed by Hu et al. [46] and the $R_s(Q_1R_{ct1})(Q_2R_{ct2})$ model proposed by Gu et al. [47], which are shown in (a) and (b) in Figure 7. Compared with the model $R_s(Q(R_{ct}W))$, the model $R_s(Q_1R_{ct1})(Q_2R_{ct2})$ further considers the electrochemical reaction related to charge transfer at the solid–liquid interface, but this model is more suitable for studying cement-based materials with low curing humidity. Dong [48] et al. studied the AC impedance test of cement-based materials in carbonation curing, and optimized the equivalent circuit $R_s(Q_1(R_{ct1}W_1))(Q_2(R_{ct2}W_2))$, as shown in Figure 7c. Considering that the internal material humidity can reach 70% after carbonation curing, and the ions' diffusion process of the solid–liquid interface cannot be ignored, the equivalent circuit model $R_s(CR_{ct1}(QR_{ct2}))$ proposed by Song [49] is used to carry out carbonation analysis of cement-based materials, as shown in Figure 7d.

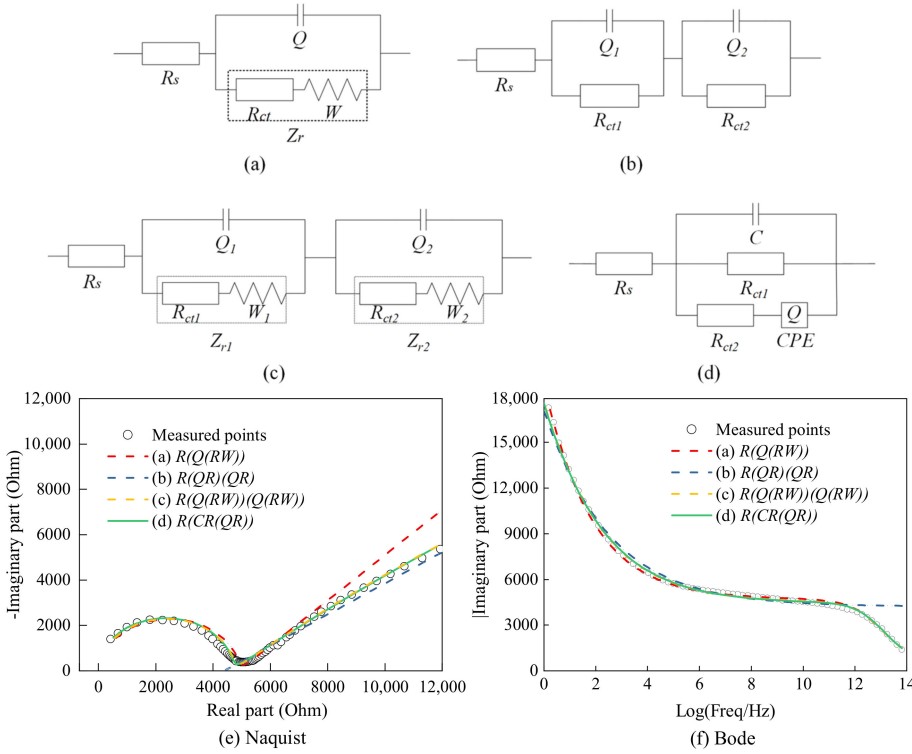

**Figure 7.** Modeling and fitting of electrochemical equivalent circuit of carbonation process: (**a**) $R_s(Q(R_{ct}W))$; (**b**) $R_s(Q_1R_{ct1})(Q_2R_{ct2})$; (**c**) $R_s(Q_1(R_{ct1}W_1))(Q_2(R_{ct2}W_2))$; (**d**) $R_s(CR_{ct1}(QR_{ct2}))$; (**e**) Nyquist; (**f**) Bode.

The comparison of fitting results of the above models is shown in Figure 7e,f; the former is a Nyquist plot, reflecting the relationship between the real part and the imaginary part of the impedance of the specimen at different frequencies, and the latter is a Bode phase diagram. The curve describes the relationship between the impedance modulus and frequency. Both of the above diagrams show that the used model had a good fitting effect.

Since the test used a three-electrode system, the compensation resistance (Weber resistance) during the EIS test could be ignored, and the high frequency arc deflection of the matrix was not obvious (the center of the circle was below the real axis). When a three-electrode system is used for measurement, the contact area between the test piece and

the voltage sensor has great influence on impedance spectroscopy. Because the semicircle in the high frequency area of the test result can pass through the origin better, the point contact test is used.

### 4.2. Analysis of Impedance Spectroscopy Parameters in Carbon Curing

The impedance spectroscopy of each specimen was fitted according to Figure 7d, and $\rho_s$, $\rho_{ct1}$, $C$, $\rho_{ct2}$, and $CPE$ could be obtained from each fitting curve. The fitting results are shown in Table 4; $\rho_s$ represents the electrolyte resistivity, which fluctuates between 0.04 and 0.20 kΩ·cm, but the regularity is not obvious. $\rho_{ct1}$ is regarded as solid phase resistivity, which is large and represents the non-conductive property of the solid phase of cement-based materials. $C$ represents the electric double-layer capacitance, showing a trend of first decreasing and then increasing, and its value was minimal after curing for 7 days and 14 days. This may be because the curing age was short and there were more free charges in the solid phase, so the capacitance was high. After curing for longer than 14 days, its microstructure was stable and dense, so the distance between free charges was small, and the capacitance was also high. $\rho_{ct2}$ represents the resistivity of charge transfer of free ions in pore fluid in cement mortar and is the main characterization parameter of cement hydration or carbonation development, corresponding to high frequency arc diameter. $\rho_{ct2}$ increased gradually with the increase of carbonation curing age, but the resistivity decreased at the third day. The decrease range for 0.4, 0.5, and 0.6 water–cement ratios were 27.5%, 51.8%, and 19.3% of $\rho_{ct2}$ at the first-day carbonation. It showed that the greater the overall resistivity, the greater the decline. For $C$, $\rho_{ct1}$, and $CPE$, the range of numerical changes was mostly in the same order of magnitude, and the order of magnitude was too large or too small, so the correlation between these parameters was relatively not obvious.

**Table 4.** Fitting result for EIS of carbonation group.

| w/c | Age | $\rho_s$ (kΩ · cm) | $C$ (kF · cm) | $\rho_{ct1}$ (kΩ · cm) | $CPE$ (kF · cm) | $\rho_{ct2}$ (kΩ · cm) |
|---|---|---|---|---|---|---|
| | 1 d | 0.158 | $4.913 \times 10^{-13}$ | $9.342 \times 10^{15}$ | $7.939 \times 10^{-8}$ | 2.913 |
| | 3 d | 0.140 | $4.628 \times 10^{-13}$ | $1.030 \times 10^{16}$ | $7.790 \times 10^{-8}$ | 2.112 |
| 0.4 | 7 d | 0.199 | $2.455 \times 10^{-13}$ | $1.273 \times 10^{16}$ | $7.955 \times 10^{-8}$ | 5.137 |
| | 14 d | 0.079 | $2.249 \times 10^{-13}$ | $1.560 \times 10^{16}$ | $1.032 \times 10^{-7}$ | 4.852 |
| | 28 d | 0.036 | $3.914 \times 10^{-13}$ | $5.553 \times 10^{16}$ | $5.773 \times 10^{-8}$ | 5.145 |
| | 56 d | 0.037 | $3.796 \times 10^{-13}$ | $5.076 \times 10^{16}$ | $3.620 \times 10^{-8}$ | 8.120 |
| | 1 d | 0.101 | $1.686 \times 10^{-13}$ | $5.264 \times 10^{13}$ | $4.082 \times 10^{-8}$ | 8.002 |
| | 3 d | 0.133 | $1.776 \times 10^{-13}$ | $1.527 \times 10^{14}$ | $6.800 \times 10^{-8}$ | 3.851 |
| 0.5 | 7 d | 0.181 | $1.464 \times 10^{-13}$ | $6.322 \times 10^{14}$ | $4.930 \times 10^{-8}$ | 8.332 |
| | 14 d | 0.081 | $1.297 \times 10^{-13}$ | $2.088 \times 10^{14}$ | $3.749 \times 10^{-8}$ | 8.440 |
| | 28 d | 0.058 | $1.804 \times 10^{-13}$ | $4.895 \times 10^{15}$ | $3.361 \times 10^{-8}$ | 10.174 |
| | 56 d | 0.043 | $2.083 \times 10^{-13}$ | $1.806 \times 10^{16}$ | $3.500 \times 10^{-8}$ | 13.340 |
| | 1 d | 0.122 | $5.767 \times 10^{-13}$ | $2.185 \times 10^{14}$ | $8.792 \times 10^{-8}$ | 2.967 |
| | 3 d | 0.094 | $5.515 \times 10^{-13}$ | $2.101 \times 10^{14}$ | $8.210 \times 10^{-8}$ | 2.394 |
| 0.6 | 7 d | 0.060 | $3.682 \times 10^{-13}$ | $4.550 \times 10^{11}$ | $6.415 \times 10^{-8}$ | 5.339 |
| | 14 d | 0.053 | $3.449 \times 10^{-13}$ | $5.679 \times 10^{15}$ | $5.343 \times 10^{-8}$ | 5.696 |
| | 28 d | 0.134 | $2.748 \times 10^{-13}$ | $6.616 \times 10^{8}$ | $4.801 \times 10^{-8}$ | 5.157 |
| | 56 d | 0.079 | $4.854 \times 10^{-13}$ | $1.129 \times 10^{16}$ | $1.608 \times 10^{-8}$ | 6.171 |

### 4.3. Electrochemical Parameter Analysis of Carbonation and Hydration Curing

The $\rho_{ct2}$ of carbon-curing specimens was obviously greater than that of hydration curing, and the $\rho_{ct2}$ of each group of three specimens was averaged, and a comparison chart was plotted as shown in Figure 8. It is not difficult to find that when the impedance modulus of the specimen was in a stable state, that is, from days 7 to 28, the impedance spectroscopy parameter $\rho_{ct2}$ of the cement-based material in carbon curing was about three times that in water curing. In addition, the $\rho_{ct2}$ of the three groups of specimens

with diverse water–cement ratios dropped sharply in carbonation curing for three days. Under the condition of water curing, $\rho_{ct2}$ of 0.4, 0.5, and 0.6 water–cement ratios increased from 1.410 k$\Omega$ · cm, 1.102 k$\Omega$ · cm, and 0.716 k$\Omega$ · cm to 2.994 k$\Omega$ · cm, 2.934 k$\Omega$ · cm, and 2.121 k$\Omega$ · cm, respectively. $\rho_{ct2}$ showed a trend of decreasing with the increase of water–cement ratio, and the resistivity of a low water–cement ratio was obviously higher than that of a high water–cement ratio.

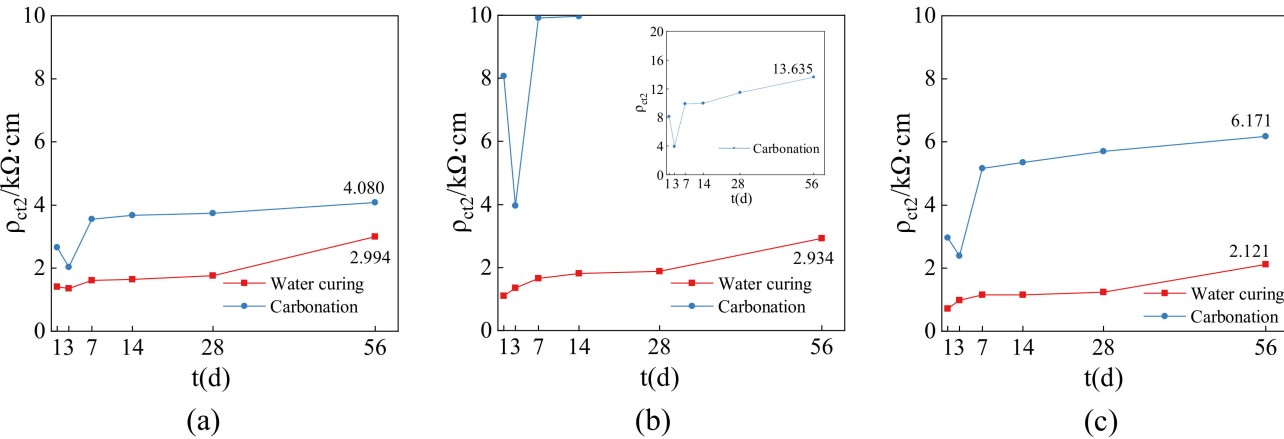

**Figure 8.** Fitted $\rho_{ct2}$ of EIS under carbonation and water curing: (**a**) w/c 0.4; (**b**) w/c 0.5; (**c**) w/c 0.6.

For the entire hydration process, at the beginning, with the continuous dissolution of cement particles, the solution inside the matrix reaches the supersaturated state, the precipitation of CH is generated in the micropores, and the ions' concentration begins to become stable. The impedance modulus continues to increase with the increase of the amount of hydration products. At the beginning of carbon curing, the resistivity decreases. This may be because the ions dissolving from the cement phase in the dissolution stage reduce the impedance modulus of the specimens until it reaches a saturated state [48]. After that, the hydration products precipitate, and the hydration enters the accelerated stage, and its impedance modulus rises rapidly. As hydration continues, the impedance modulus of the specimen is basically stable between days 7 and 28, the cement hydration is gradually in the dynamic equilibrium stage, and the impedance modulus increases significantly again between days 28 and 56. It may be that the hydration products tend to develop into crystals in a more stable state.

### 4.4. Relationship between Carbonation and Electrochemical Parameters

Cement-based materials have remarkable porous properties, and their pore structure exhibits certain fractal characteristics [50]. In order to study the evolution of the pore structure of cement-based materials by the impedance method, it is necessary to explain the "penetration" ability of the AC signals emitted by the workstation in the pore structure by simulating the circuit parameters into the internal pore structure of the working electrode, that is, the specimen. The electrical transmission types of AC signals in porous media can be divided into three categories, as shown in Figure 9. The principle of pore structure network shows cement matrix, continuous pore, and discontinuous pore. In fact, pore structure can also be classified by pore size [51].

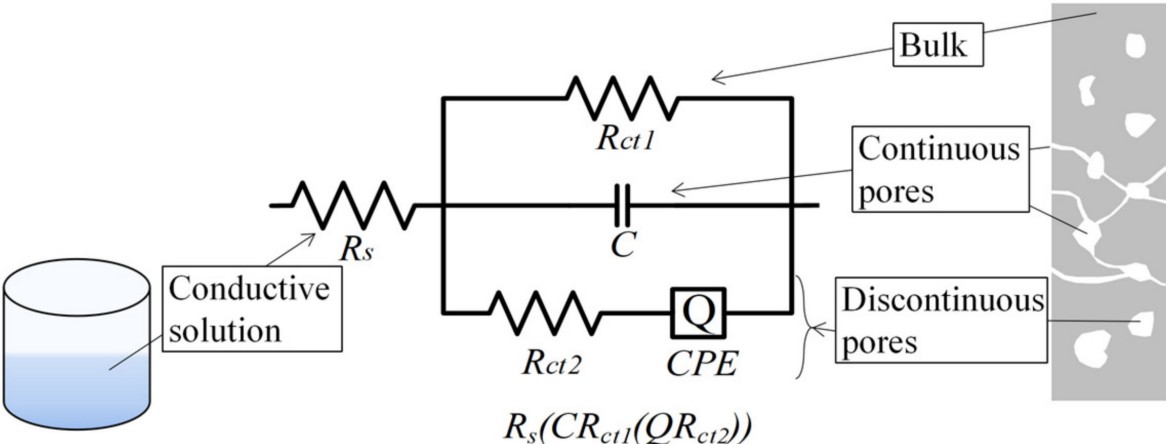

**Figure 9.** Relationship between equivalent circuit and microstructure.

Based on the characteristics of the pore structure network and electrical response characteristics, it is assumed that the solid phase part of cement-based materials is explained by $R_{ct1}$ of the maximal resistance in the equivalent circuit. Holes with different apertures meet the electrical response of double-layer capacitance in the equivalent circuit. Each hole is assumed to be a cylinder with a smooth inner wall, and its interior is filled with an electrolyte solution. In addition, the closed cell meets the electrical response of the joint action of resistance $R_{ct2}$ and capacitance $CPE$, representing the number and aperture of closed cells in the cement-based material pore structure. The electrolyte provides the initial ion concentration to meet the electrical response of resistance $R_s$ in the equivalent circuit.

According to Fick's second law, the penetration rate of carbonation is in direct proportion to $\sqrt{t}$ [52], and the impedance spectroscopy fitting parameter $\rho_{ct2}$ changes significantly in carbonation, so this study selects $\rho_{ct2}$ to establish a relationship with carbonation rate, and the relationship law to be established is as follows:

$$Cr = a(\rho_{ct2} - b)^{0.5} \tag{4}$$

where $Cr$ is the carbonation rate, %, and $a$ and $b$ are the fitting parameters. Assuming a suitable value of $b$, taking $(\rho_{ct2} - b)^{0.5}$ as an unknown part, the straight line is fitted according to the linear law, and when the correlation coefficient $R^2$ is large, the fitting effect is excellent. After several trials, the results show that when the water–cement ratio was 0.4, when $b$ takes 1.570, the fitting effect was better, and the correlation coefficient was 0.9511 at this time, the value of $a$ was 1.551, as shown in Figure 10a. Similarly, the carbonation rate and $\rho_{ct2}$ at the water–cement ratio of 0.5 and 0.6 were fitted, and the relationship law shown in Figure 10b and c was obtained. It was found that when the water–cement ratio was 0.6, the fitting effect was the best and the correlation coefficient reached 0.9857. The relationship model between the carbonation mass increase rate and the impedance spectroscopy parameter $\rho_{ct2}$ was established. The carbon sequestration amount of carbon-curing cement-based materials with different water–cement ratios could be predicted by resistivity in the electrochemical AC impedance analysis test, and the verification and prediction of concrete carbon curing could be realized by electrochemical AC impedance technology.

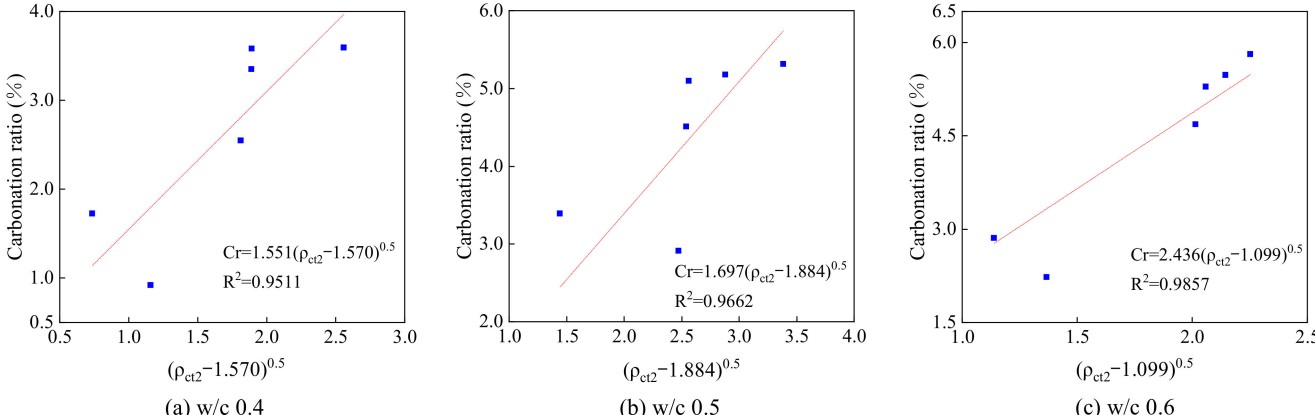

**Figure 10.** Relationship between $\rho_{ct2}$ and carbonation ratio: (**a**) w/c 0.4; (**b**) w/c 0.5; (**c**) w/c 0.6.

## 5. Conclusions

Electrochemical impedance of cement-based materials in the carbon-curing process showed regular changes, the arc diameter in the high frequency region showed an increasing trend, and the corresponding resistivity parameter $\rho_{ct2}$ increased significantly.

When the impedance modulus of the specimen was in a stable state between days 7 and 28, the resistivity $\rho_{ct2}$ of cement mortar with carbonation curing was about three times that of the water curing. Since the microstructure of cement mortar in the early age had not been fully formed, there were plenty of free ions in the pores. With the gradual accumulation of hydration products in the long-term age, the $CaCO_3$ content accumulated with the interaction of carbonation process, thus the microstructure of cement mortar was gradually compacted.

The relationship model between the mass increased rate in the carbonation process and the impedance spectroscopy parameter $\rho_{ct2}$ was established, and the diffusion process of $CO_2$ in mortar followed Fick's law, and both of them were quadratic functions. Therefore, the monitoring of concrete carbonation could be realized by EIS measurement.

In the future, the degree of carbonation can be studied in combination with TG and EIS measurement to characterize the microstructure mechanism of the carbonation process. The electrochemical impedance spectroscopy analysis can be applied to monitor the carbon capturing of cement-based materials and their durability, either for newly mixed concrete or for recycled concrete materials.

**Author Contributions:** Conceptualization, Q.L. and H.T.; methodology, Q.L. and H.T.; investigation, Q.L., H.T. and L.C.; data curation, H.T.; writing—original draft preparation, Q.L. and H.T.; writing—review and editing, L.C., K.C., L.Z. and C.L. All authors have read and agreed to the published version of the manuscript.

**Funding:** Jingxiong Project: Research on Rapid Verification Technology of Concrete Materials and Quality Defects of Bridges and Culverts Under Construction (JX-202018).

**Institutional Review Board Statement:** Not applicable.

**Informed Consent Statement:** Not applicable.

**Data Availability Statement:** The data presented in this study are available on request from the corresponding author. The data are not publicly available due to business restrictions.

**Conflicts of Interest:** The authors declare no conflict of interest.

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
