# Peer review of "Evolution of Electrochemical Impedance Spectra Characteristics of Cementitious Materials after Capturing Carbon Dioxide"

_sustainability, doi:10.3390/su15032460_

Round 1

Reviewer 1 Report

Attached.

Author Response

General comment

The paper “Evolution of electrochemical impedance spectra characteristics of cementitious materials after capturing carbon dioxide” submitted by Qiong Liu et al. could be accepted after the following minor revision.

Response:

Thank you for your kindly comments on our manuscript.

Comment 1

The given abstract is mostly based on general discussion, the authors should discuss the main outcomes of their study with some outstanding results. The general part could be shifted to the introduction.

Response:

Thank you for your suggestion. The abstract has been revised to enclose more useful results and analysis.  

Comment 2

The authors should explain the importance of electrochemistry in cement and construction-based materials such as given in the following paper

(https://doi.org/10.1002/tcr.202200134).;

Response:

Thank you for your comments. The introduction has been refined according to the reviewer’s advice and the related reference has been added to explain the importance of electrochemistry in cement-based materials.

Comment 3

The main focus of this paper is based on EIS, but the introduction part does not explain the EIS properly. The authors should mention the various frequency ranges and their explanation like charge transfer resistance and Warburg impedance, by reading and citing some recent literature such as https://doi.org/10.1016/j.est.2022.104278.

Response:

Thank you for your useful and kindly comments on the introduction of our manuscript. The introduction has been revised according to the reviewer’s advice and this research has been added in appropriate sections.

Comment 4

The authors should provide an equivalent circuit diagram for their electrochemical cell used in EIS measurements in Figure 2 and Figure 3.

Response:

Thanks for your comments, the equivalent circuit diagram has been added in the Figure 2(a) and the reason why we select this diagram has been explain in appropriate sections.

Comment 5

Some physical characterizations of the prepared sample are needed for a proper understanding of the material’s behavior. It is recommended to provide SEM and XRD

data before and after curing of the prepared specimen.

Response:

Thanks for your advices, the new discuss section has been created, including the XRD patterns, in the 3.4 section.

Comment 6

As shown in Figure 2 and Figure 3, the solution resistance remain the same even after 56d, how this could be possible as some of the solution could block the accessible surface area and will increase the resistance. This has to be explained with a proper evidence.

Response:

Thank you for your useful and kindly comment, the new analysis has been added in appropriate sections.

Reviewer 2 Report

The paper "Evolution of electrochemical impedance spectra characteristics of cementitious materials after capturing carbon dioxide" has showed a thorough analysis of the changes in the electrochemical impedance spectra (EIS) of cementitious materials after capturing CO2. I think the paper presents a comprehensive study of the effect of CO2 capture on the EIS of cementitious materials and provides valuable insights into the underlying mechanisms of the carbonation process. The results of the study have also revealed the durability and long-term performance of cement-based materials in carbon capture and storage systems.

I appreciate the use of multiple EIS analysis methods and good data delivery in the paper, which allows for a more comprehensive understanding of the changes in the cementitious materials' EIS characteristics.

One minor issue is that the authors could have provided more context on the significance of the changes in the EIS characteristics in relation to the carbonation process and the potential impacts on the durability and performance of the cementitious materials.

Overall, this is a well-conducted study that provides useful insights into the changes in EIS characteristics of cementitious materials after capturing CO2.

Author Response

Comment

One minor issue is that the authors could have provided more context on the significance of the changes in the EIS characteristics in relation to the carbonation process and the potential impacts on the durability and performance of the cementitious materials.

Response:

Thank you for your kindly comments on our manuscript. The whole manuscript has been reviewed and some parts has been revised by considering the reviewer’s suggestion on the context of the EIS characteristics in relation to the carbonation process and the potential impacts on the durability and performance of the cementitious materials.

Reviewer 3 Report

Its well written and research is good in this field. Few english discripencies are there, please correct those, some are mentioned in Marked Manuscrit. Minor revision for English and editing.

Author Response

Comment

The reviewer marked some grammatical errors and incongruous sentences in the manuscript.

Response:

Thank you for your kindly comments on our manuscript. All the mistakes and inappropriate statements have been revised, and the authors have checked grammatical errors and statements throughout the manuscript. Main modifications are identified in blue font.